# LS-CLIP: AUTOENCODER-BASED MINING OF CLIP'S INHERENT LOCAL SEMANTICS IN CROSS-DOMAIN IMAGE RETRIEVAL

## ABSTRACT

Contrastive Language-Image Pretraining (CLIP) excels in image retrieval. However, existing methods often depend on extensive manual annotations for local supervision and ignore CLIP's native local-semantic capabilities. To address these problems, we propose an autoencoder-based approach named LS-CLIP, which is designed to extract local semantics in CLIP and realize cross-domain feature alignment. First, we design a self-supervised Semantic Reconstruction Module (SRM) for local feature mining. By adding reconstructing the patch features of the Vision Transformer (ViT) to class token features, SRM integrates global and local semantic information to adapt to retrieval tasks of different granularities. Second, To enhance the model's ability to generalize across different domains, we introduce Feature Moment Transfer (FMT). Through the reconstruction of cross-domain features via moment (mean and variance) transfer, the stability of the feature space is enhanced. In addition, this module incorporates noise when reconstructing the data distribution, thereby improving the generalizability of the model. To accommodate diverse retrieval intents, we construct a dataset with rich textual descriptions and a wide range of scenarios, named CDIR-Flickr30k. Extensive experiments demonstrate that LS-CLIP significantly outperforms SOTA baseline models in various metrics. Zero-shot evaluation confirms its strong generalizability. Importantly, LS-CLIP can be applied as a plug-and-play model to CLIP variants, consistently delivering performance improvements.

## 1 INTRODUCTION

Query-Based Image Retrieval (QBIR) Datta et al. (2008); Thomee & Lew (2012); Isinkaye et al. (2015); Li et al. (2023a) is the task of retrieving relevant images from a large image database using user queries or search terms. With the development of this field, challenges have expanded from single-domain retrieval to Cross-Domain Image Retrieval (CDIR) Ghosh et al. (2018); Wang et al. (2017). CDIR means to find relevant images in one visual domain based on query images from another visual domain, such as sketches, paintings, or photographs Wang et al. (2014).

In practical scenarios, users often express retrieval needs via various query formats. However, existing retrieval research exhibits clear limitations: QBIR predominantly focuses on text queries, while CDIR centers on cross-domain image-to-image retrieval. This significantly restricts the range of query types available to users. In existing datasets, FSCOCO Chowdhury et al. (2022) supports image retrieval of text or sketch queries. In CDIR the DomainNet Peng et al. (2019) dataset consists of 6 domains, which are collected from multiple sources. FreestyleRet Li et al. (2024) expands to cartoons and low-quality images. There are no datasets that connect to retrieving a full image via a local detail image, which is a common real-world need. To enrich the evaluation dimensions and broaden the assessment perspectives, we propose a versatile dataset named CDIR-Flickr30k, which is based on Flickr30k Young et al. (2014). All textual descriptions are completed by professional annotators with comprehensive semantic coverage. Additionally, the images are sourced from real user uploads, featuring natural scenarios and rich semantic information. Additionally, we have extended image to sketch, cartoon, low quality, and object retrieval tasks, using existing technologies. This dataset enables comprehensive validation of the performance of our method in diverse scenarios. It

Figure 1: Comparison of attention response heatmaps among the CLS token, patch tokens, and our proposed all-patch tokens after global pooling. The green dotted box denotes the CLS token, the yellow dotted box denotes patch tokens, and the red dotted box denotes our all-patch tokens. △ is a image patch randomly selected on the target object. More results can be seen in appendix A.4.

supports text-based image retrieval, adapts to cross-domain image-to-image retrieval, and fulfills the requirements of image-to-image retrieval at varying granularities.

Whether in text-to-image or image-to-image retrieval tasks, acquiring fine-grained information significantly impacts retrieval performance. This problem is critical for image retrieval as cross-domain images often share semantic equivalence but differ in local visual forms. The local semantic deficiency further causes the domain-gap issue, making image retrieval methods still far from practical application. To enhance the ability of local semantic understanding, existing CLIP-based methods have explored targeted improvements. For example, RO-ViT Kim et al. (2023) adopts a contrastive approach to align sentence tokens with image regions, while FG-CLIP Xie et al. (2025) uses region-text pairs for enhanced regional contrastive learning. Despite their progress, these methods share two key limitations. Firstly, they are highly dependent on supervised learning. This requires large-scale manual annotations, such as object bounding boxes and region-text alignments. These annotations increase data costs and at the same time limit their scalability. Secondly, they overlook the inherent local semantic potential of the CLIP Vision Transformer (ViT) structure, specifically, patch features that naturally encode local object information.

To verify this potential, we analyzed CLIP attention response heatmaps as shown in Fig.1. The CLS token focuses mainly on global semantics. The CLS of the low-level layer has more attention to local details but more focuses on the background. In contrast, ViT patch features exhibit strong targeted attention to different objects, providing valuable supplementary local semantic for cross-domain alignment. The self-attention-based MAE (Masked Autoencoder) He et al. (2022) method excels at capturing image details. Its core idea lies in leveraging reconstruction loss to enable the model to learn effective image representations for subsequent downstream tasks. Inspired by this observation, we designed an autoencoder-based adapter named Semantic Reconstruction Module (SRM) to extract local semantics from patch features. By combining this local information with the global semantics of the CLS token, SRM can address both the issues of high supervision dependence and insufficient utilization of CLIP's local semantic extraction capability in existing methods. Unlike MAE, the proposed SRM avoids the loss of critical visual details during compression by reconstructing the original image patch features after compressing them. Furthermore, the SRM decoder is only used for auxiliary training and is not invoked during the model inference phase. Furthermore, to improve the stability of feature spaces and generalization, we propose Feature Moment Transfer (FMT). Inspired by the AdaIN Huang & Belongie (2017) style transfer framework, FMT optimizes the alignment of feature distribution across domains and incorporates controlled noise to reduce overfitting to specific domains. Additionally, the FMT module in our framework serve as auxiliary constraint modules during the training phase. As shown in Fig.1, LS-CLIP forcing the model to learn the semantic associations of local regions. As a result, the attention automatically focuses on the core local features of the object.

In summary, our contributions are as follows.

- We propose LS-CLIP, a lightweight and flexibly pluggable feature enhancement framework for image retrieval. Integrates the Semantic Reconstruction Module (SRM) and Feature Moment Transfer (FMT) to achieve a more accurate understanding of both local and global semantic information, addressing key limitations of existing CLIP-based methods.

- We construct the CDIR-Flickr30k dataset, which contains rich textual descriptions and an extensive scenario. This dataset supports the comprehensive validation of cross-domain retrieval methods, including multigranularity retrieval tasks.

- Extensive experiments demonstrate that LS-CLIP outperforms state-of-the-art baseline models on both our CDIR-Flickr30k dataset and other public retrieval datasets. It also exhibits strong generalization capability through zero-shot evaluation, providing effective solutions and new insights for practical image retrieval applications.

## 2 RELATED WORK

**Image Retrieval** CDIR further increases image retrieval challenge by incorporating search tasks across different domains such as sketches, cartoons, paintings, and photographs Datta et al. (2008); Huang et al. (2015). There is a significant visual domain gap between queries and targets in CDIR. It leads to misalignment of feature distributions and degraded retrieval performance. Early approaches leveraged category information for discriminative feature extraction or minimized losses such as triplet Yu et al. (2016) and HOLEF Song et al. (2017) for cross-domain pairing. However, these methods have a limitation. They show poor generalization across domain pairs. For example, methods optimized for sketch-photo fail on cartoon-photo tasks. The difference is that in the QBIR field, with the development of large-scale VLMs, such as, CLIP Radford et al. (2021), ALIGN Jia et al. (2021), BLIP-2 Li et al. (2023b), cross-modal alignment is used in QBIR. These models leverage pretrained image-text semantic associations to bridge domain gaps, allowing tasks such as text-image retrieval Li et al. (2022a); Radford et al. (2021) and text-video retrieval Jin et al. (2023a;b). Most VLMs-based QBIR methods only use the CLS token ignoring local details critical to cross-domain matching. This problem causes performance drops in the search task for full image using the local object query.

**Datasets** Image-text datasets are the foundation of VLMs-based QBIR. However, existing benchmarks have incomplete coverage of the retrieval scenario. Datasets like LAION Jia et al. (2021), COCO Lin et al. (2014), and Flickr30K Young et al. (2014) focus on contrastive text-image learning. Diverse-Style Retrieval Dataset (DSR) Dataset Li et al. (2024) extend to multistyle domains but lack local object retrieval, a common real-world scenario. This gap makes it impossible to assess the ability of the method to adapt to multigranularity CDIR, which motivates us to construct the CDIR-Flickr30k dataset.

**Local Semantic Enhancement** Most VLMs (e.g. CLIP Radford et al. (2021)) have limited local semantics due to be optimized on global image-text alignment, which hampers specific region feature extraction. One kind of enhancement method is to depend on supervision. For example, GLIP Li et al. (2022b), RegionCLIP Zhong et al. (2022) use grounding data which need labor-heavy annotations. FG-CLIP Xie et al. (2025) relies on object captions and ignores ViT patch features. In addition, some methods are only based on simple image-text pairs. LongCLIP Zhang et al. (2024) extends the length of the text. However, it ignores the local information from the VLMs. In summary, existing methods have two flaws. Firstly, high reliance on annotations or one-sided text optimization. Secondly, ignore the local semantics of ViT. We propose SRM to mine local semantics from ViT patches via self-supervised reconstruction without extra annotations.

**Autoencoder** The autoencoder He et al. (2022); Hou et al. (2022); Wei et al. (2023) enables unsupervised learning via encoder-decoder structure by minimizing input-output reconstruction error, reducing labeled data reliance. It shows promise in CDIR but has limitations in existing applications. CDFD He (2024), which is an unsupervised CDIR, uses DWT and DCT for robustness. However, it relies on hand-crafted transforms and fails to capture high-level semantics across domains, which limits semantic-driven CDIR. CM Iijima et al. (2024) uses VLMs to generate captions as CDIR intermediates to avoid using labeled data. However, it depends on caption quality. For example, ambiguity hurts alignment. It also increases the computation cost of caption generation. Our work uses autoencoder-based unsupervised reconstruction to address those problems.

## 3 METHODOLOGY

In this section, we first present an overview of our proposed LS-CLIP, which contains the Semantic Reconstruction Module (SRM) and the Feature Moment Transfer (FMT). Next, we elaborate on the

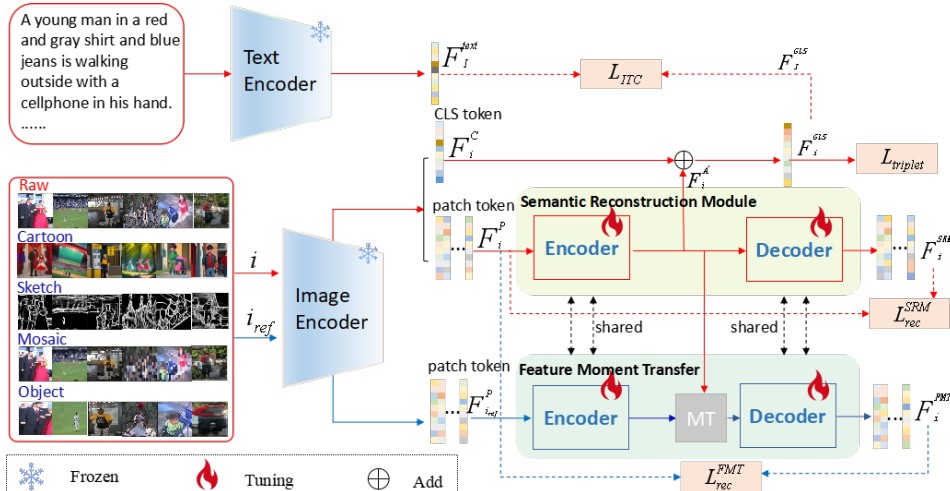

Figure 2: The Overall Framework of our LS-CLIP. The model input $i$ denotes an image from $[I, I_{pos}, I_{neg}]$, and the input $i_{ref}$ is an randomly selected corresponding domain images to $i$. We propose a Semantic Reconstruction Module (SRM) based on autoencoder architecture, which is used to mine local semantic information from patch features. Additionally, through the Feature Moment Transfer (FMT), we perturb the feature distribution and perform reconstruction to achieve the stability of the feature space. $L_{ITC}$ is the InfoNCE loss in Equation (5), $L_{triplet}$ is the triplet loss to achieve modality alignment in Equation (6), $L_{rec}^{SRM}$ and $L_{rec}^{FMT}$ is MSE loss to mine local semantics in Equation (3) and (8). $F_i^{GLS}$ is feature of the input $i$ for retrieval.

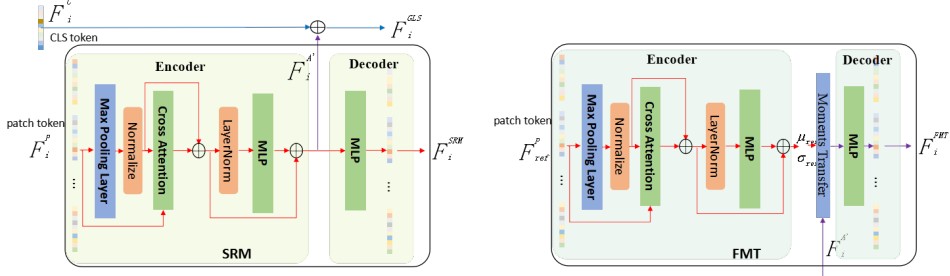

Figure 3: Model architecture of LS-CLIP with SRM and FMT. The encoder is a multi-head cross-attention block with a Max Pooling Layer to pool the image features at first. For SRM and FMT, they share the same model parameters.

detailed network architecture, with a particular focus on SRM and FMT. Finally, we introduce the retrieval tasks designed to validate the effectiveness and generalization of our proposed LS-CLIP.

## 3.1 NETWORK ARCHITECTURE

### 3.1.1 SEMANTIC RECONSTRUCTION MODULE

To extract the local semantic information of CLIP, we introduce the Semantic Reconstruction Module (SRM), which is a lightweight module. SRM reconstructs patch features into low-dimensional representations that preserve local semantics, then integrates these representations into the CLS token to enhance the ability of local semantic understanding. The SRM encoder consists of a Max-Pooling Layer and a Multi-head Cross-Attention Layer which as shown in left of Fig.3. For an anchor image $I$, it will be inputted into the frozen CLIP image encoder to extract feature $F_I \in R^{(1+n) \times d}$, where $n$ denotes the number of ViT patch tokens in CLIP and $d$ denotes the dimension of feature. $F_I$ is formulated as $F_I = [F_I^C | F_I^P]$ (where $[|]$ denotes the concatenation operation), in which $F_I^C \in R^{1 \times d}$ and $F_I^P \in R^{n \times d}$ are the CLS features and patch features, respectively. To optimize triplet loss, we select two auxiliary samples. One is a negative sample $I_{neg}$ which is a semantically irrelevant natural image from the same dataset as $I$. One is a positive sample $I_{pos}$ which

is a semantically consistent cross-domain image, such as sketch corresponding to $I$. In the same way, we extract their features $F_{I_{neg}} \in R^{(1+n) \times d}$ and $F_{I_{pos}} \in R^{(1+n) \times d}$. $F_I$, $F_{I_{neg}}$ and $F_{I_{pos}}$ are further used in SRM for local semantic reconstruction.

**Max Pooling Layer**. For patch features $F^P$ of any image, it first undergoes a maximum pooling operation for dimensionality reduction to obtain $F^{P'}$, that is, $F^{P'} = Pool(F^P)$. The purposes of this step are as follows. First, it can compress useful information to minimize interference from redundant information. Second, it can ensure that $F^{P'}$ and the CLS token features $F^C$ maintain consistent feature scales, facilitating subsequent operations such as feature fusion.

**Multi-head Cross-Attention Layer**. After compression of the information through the pooling operation, there may be information distortion in $F^{P'}$. Therefore, a multi-head cross-attention layer is introduced here to allow information interaction between $F^P$ and $F^{P'}$, as shown in the formula (1). The significant advantage of this approach is that we do not change the dimension of the feature and the number of scales of $F^{P'}$ itself in this process, which means that its maximum information carrying capacity remains fixed. Through the interaction of information with $F^P$, the model can compress more modal information into $F^A$. After that, $F^A$ will be fed into a light MLP network to obtain $F^{A'}$, which can be formulated as equation (2). Finally, the fusion of the features of $F^C$ and $F^{A'}$ produces the final feature of the image $F^{GLS}$, which can be formulated as $F^{GLS} = F^C + F^{A'}$. In the inference phase, we will use $F^{GLS}$ as the semantic feature of the image for CDIR.

$$F^A = \textbf{Attention}(Q, K, V, F^{P'}, F^P) = \textbf{softmax}(\frac{Q(F^{P'})K(F^P)}{\sqrt{d_k}})V(F^P) \tag{1}$$

$$F^{A'} = \textbf{MLP}(F^A + F^{P'}) + (F^A + F^{P'}) \tag{2}$$

In the decoder, we choose to adopt a lightweight MLP network for semantic reconstruction. The feature $F^{A'}$ is input into the decoder to obtain $F^{rec} = \textbf{D}(F^{A'})$.

The autoencoder is independent of any specific image domain and does not require labeled data. For any image $i$, we can get their reconstruction results $F_i^{rec}$ through SRM. We employ MSE loss as an optimization objective for reconstruction. $L_i^{SRM}$ is computed between $F^{rec}$ and $F^P$ as in Equation (3). The final loss of reconstruction $\mathcal{L}_{rec}^{SRM}$ is denoted as the mean of $\mathcal{L}_i^{SRM}$ for all images.

$$\mathcal{L}_i^{SRM} = (\sum_{n=1}^{b} \sum_{j=1}^{hw} \sum_{k=1}^{d} (F_{i,n,j,k}^{rec} - F_{i,n,j,k}^{P})^2)/(b \times hw \times d) \tag{3}$$

Furthermore, to preserve the original image-text alignment capability of the model, we retain the original image-text contrastive loss $\mathcal{L}_{ITC}$ as an optimization objective. To maintain semantic consistency, we only compute the InfoNCE loss Oord et al. (2018) for the nature image $I$. Specifically, $\mathcal{L}_{ITC}$ can be expressed as in equation (5), where $\mathcal{L}_{i2t}^I$ and $\mathcal{L}_{t2i}^I$ are defined as equation (4). $\delta(.)$ is denoted as the distance of cosine similarity in equation (4).

$$\mathcal{L}_{i2t}^I = -log \frac{exp(\delta(F_I^{LS-CLIP}, F_I^{text})/\tau)}{\sum_{k=1}^{N} exp(\delta(F_I^{LS-CLIP}, F_k^{text})/\tau)} \tag{4a}$$

$$\mathcal{L}_{t2i}^I = -log \frac{exp(\delta(F_I^{LS-CLIP}, F_I^{text})/\tau)}{\sum_{k=1}^{N} exp(\delta(F_k^{LS-CLIP}, F_I^{text})/\tau)} \tag{4b}$$

$$\mathcal{L}_{ITC} = (\mathcal{L}_{i2t}^I + \mathcal{L}_{t2i}^I)/2 \tag{5}$$

Finally, to align semantic feature spaces between cross-domain images, we utilize triplet loss $\mathcal{L}_{triplet}$ as an optimization objective. Our triplet loss objective is to fully learn the semantic information between cross-domain images by reducing the distance between image $I$ and the positive sample $I_{pos}$ while increasing the distance between image $I$ and the negative sample $I_{neg}$, which can be denoted by Equation (6) with margin $\gamma > 0$. Triplet loss aims to minimize the distance $\delta(.)$ between $F_I^{GLS}$ and $F_{I_{pos}}^{GLS}$, while increasing that of a random negative feature $F_{I_{neg}}^{GLS}$.

$$\mathcal{L}_{triplet} = \textbf{max}(0, \gamma + \delta(F_I^{GLS}, F_{I_{pos}}^{GLS}) - \delta(F_I^{GLS}, F_{I_{neg}}^{GLS})) \tag{6}$$

### 3.1.2 FEATURE MOMENT TRANSFER

To improve the generalization of the model across domains, we introduce the FMT mechanism as an auxiliary training strategy, which is not used in inference. Using feature space regularization (e.g., additive noise) merely perturbs the feature distribution without aligning the statistical properties of cross-domain data. Inspired by style transfer methods Huang & Belongie (2017), which in feature moments are used to adjust feature distributions, we use moment transfer to improve the robustness of the feature space. In the FMT, we first modify the feature distribution of the anchor image by replacing the feature moment, then reconstruct the perturbed features back to their original distribution. FMT not only enriches the diversity of feature representations, but also strengthens the adaptability of model in cross-domain retrieval scenarios.

In FMT, we first randomly select a reference image from the same domain as $i_{ref}$ but with generally inconsistent semantics compared to the image $i$. The reference image is then fed into the encoder module (which shares parameters with SRM as shown in Fig.3) to obtain its $F_{i_{ref}}^{A'}$. We replace the corresponding feature moments of the source image to complete the feature moment transfer operation. The FMT result feature $F_i^{FMT}$ of image $i$ is defined as:

$$F_i^{FMT} = \sigma_{ref}(F_i^{A'} - \mu)/\sigma + \mu_{ref} \tag{7}$$

Where $\mu_{ref}$ and $\sigma_{ref}$ is the first-order feature moment and the second-order feature moment of $F_{i_{ref}}^{A'}$. In addition, $\mu$ and $\sigma$ denote the feature moments of $F_i^{A'}$.

Subsequently, we input $F_i^{FMT}$ into the Decoder (which shares parameters with SRM as shown in Fig.3) for feature reconstruction. We perform MSE loss $\mathcal{L}_i^{FMT}$ alignment optimization with $F^P$ of the source image, which is similar to $\mathcal{L}_{rec}$ as the equation (8). The final FMT loss $\mathcal{L}_{rec}^{FMT}$ is denoted as the mean of $\mathcal{L}_i^{FMT}$ for all images $i$.

$$\mathcal{L}_i^{FMT} = (\sum_{n=1}^{b}\sum_{j=1}^{hw}\sum_{k=1}^{d}(\mathbf{D}(F_{i,n,j,k}^{FMT}) - F_{i,n,j,k}^{P})^2)/(b \times hw \times d) \tag{8}$$

Thus, the training loss function in our method is $\mathcal{L} = \mathcal{L}_{rec}^{SRM} + \mathcal{L}_{triplet} + \mathcal{L}_{ITC} + \mathcal{L}_{rec}^{FMT}$.

## 4 EXPERIMENT

### 4.1 EXPERIMENT SETTING

**Compting Methods.** To better validate LS-CLIP, we conducted image retrieval experiments to compare the model with state-of-the-art multimodal models, such as FreestyleRet Li et al. (2024), FG-CLIPXie et al. (2025), Siglip2 Tschannen et al. (2025). We utilize ViT-L-14 Radford et al. (2021) as the default vision encoder in our experiments.

**Training Set.** Four A800 GPUs (80G) are used for the training phase. The learning rate is set as the cosine warm-up learning rate. The initial learning rate is set as 1e-6, while the maximum learning rate is set as 1e-4. The batch size is 512. We train for a total of 64 epochs in our training phase.

**Evaluation Set.** The models are evaluated by recall at k (R@k) and mean average precision at k (mAP@k). For the DSR dataset and CDIR-Flickr30k, we use the open-source code of FreestyleRet Li et al. (2024) as the evaluation code with a batch size of 24.

### 4.2 DATASETS

To address the limitation of incomplete retrieval types in existing CDIR datasets, we extend the Flickr30k Young et al. (2014) dataset to construct Cross-Domain Image Retrieval Flickr30k (CDIR-Flickr30k). First, this dataset retains Flickr30k's 31,783 natural images as the target retrieval corpus and its human-annotated captions which comprehensively describe image content. Then we add four cross-domain query types to cover typical CDIR scenarios in CDIR-Flickr30k. The details of each query type are as follows. Text: we select the first caption in Flickr30k as the text prompt.

Table 1: Results (%) comparison between our method (+Ours) and other baselines on DSR. * means the model of fine-tuning on the dataset while ˆ means the model of zero-shot on the dataset.

| Methods | T → I | | S → I | | C → I | | M → I | | Average | |
|---|---|---|---|---|---|---|---|---|---|---|
| | R@1 | mAP@10 | R@1 | mAP@10 | R@1 | mAP@10 | R@1 | mAP@10 | R@1 | mAP@10 |
| FreestyleRet | 76.6 | 87.0 | 70.7 | 81.9 | 63.4 | 78.2 | 83.9 | 91.6 | 73.7 | 84.7 |
| CLIP* | 80.7 | 89.0 | 34.9 | 50.0 | 51.7 | 66.2 | 64.2 | 74.0 | 57.9 | 69.7 |
| CLIP+Ours | 82.4 | 89.4 | **79.1** | **86.6** | **68.3** | 80.6 | **93.8** | **96.3** | 80.9 | 88.3 |
| FG-CLIP* | **84.8** | **91.5** | 41.8 | 56.4 | 62.1 | 75.6 | 68.6 | 78.0 | 64.3 | 75.4 |
| FG-CLIP+Ours | 82.8 | 90.0 | 69.3 | 78.4 | 65.4 | 73.6 | 90.8 | 95.0 | 77.1 | 84.2 |
| Siglip2* | 76.7 | 85.0 | 60.3 | 72.7 | 62.9 | 76.7 | 73.3 | 82.0 | 68.3 | 79.1 |
| Siglip2+Ours | 76.7 | 85.8 | 70.9 | 86.3 | 62.9 | **81.7** | 81.4 | 90.4 | 73.0 | 86.0 |

Table 2: Results (%) of our method (+Ours) and other baselines on CDIR-Flickr30k dataset.

| Methods | T→I | | S→I | | C→I | | M→I | | O→I | | Average | |
|---|---|---|---|---|---|---|---|---|---|---|---|---|
| | R@1 | mAP@10 | R@1 | mAP@10 | R@1 | mAP@10 | R@1 | mAP@10 | R@1 | mAP@10 | R@1 | mAP@10 |
| FreestyleRetˆ | 94.7 | 97.4 | 55.3 | 72.6 | 65.0 | 75.8 | 59.0 | 78.7 | / | / | 68.5 | 81.1 |
| CLIP+Oursˆ | 96.5 | 98.1 | 81.8 | 88.0 | 79.0 | 86.6 | 83.6 | 89.1 | / | / | 85.2 | 90.5 |
| CLIP* | 97.8 | 98.8 | 46.3 | 58.5 | 72.3 | 81.6 | 51.0 | 62.6 | 95.8 | 96.9 | 72.6 | 79.7 |
| CLIP+Ours | 97.9 | 98.9 | 87.1 | 92.0 | 81.7 | **88.7** | 86.4 | 91.3 | 98.9 | 99.3 | 88.4 | 94.0 |
| FG-CLIP* | **99.0** | **99.5** | 48.3 | 62.3 | 81.5 | 87.9 | 46.8 | 60.6 | 98.1 | 98.8 | 74.7 | 81.8 |
| FG-CLIP+Ours | 98.9 | 99.4 | **89.7** | **94.1** | 81.1 | 83.9 | **87.2** | **91.9** | 99.2 | 98.6 | 91.0 | 93.6 |
| Siglip2* | **99.0** | 99.0 | 70.3 | 63.7 | **82.8** | 85.5 | 53.2 | 57.6 | 99.1 | 99.4 | 80.9 | 81.0 |
| Siglip2+Ours | 98.6 | 99.3 | 85.7 | 91.0 | 81.8 | 88.5 | 70.3 | 80.7 | **99.5** | **99.7** | 87.2 | 91.8 |

Sketch: The Pidinet Su et al. (2021) method was used to generate the sketch. Mosaic: The original images were converted into low-resolution images via downsampling as queries. Object: We use GroundingDINO Liu et al. (2024) to extract local target images based on caption descriptions.

Details of CDIR-Flickr30k are in the Appendix B. Furthermore, we adopted the DSR Li et al. (2024) and the CDIR-Flickr30k dataset as training and testing datasets.

## 4.3 MAIN RESULT

### 4.3.1 COMPARISONS ON CROSS-DOMAIN IMAGE RETRIEVAL TASK

To validate the effectiveness of the proposed LS-CLIP for CDIR tasks, we evaluate on the DSR and the CDIR-Flickr30k datasets. The results are shown in Table 1 and Table 2, respectively. We use T → I for Text to Image, S → I for Sketch to Image, C → I for Cartoon to Image, M → I for Mosaic to Image, and O → I for Object to Image. For benchmark models, such as CLIP Radford et al. (2021), Siglip2 Tschannen et al. (2025), FG-CLIP Xie et al. (2025), the incorporation of LS-CLIP has led to better performance in multiple scenarios with cross-domain queries. The results demonstrate that the integration of LS-CLIP into the three baseline models leads to significant improvements in both R@1 and mAP@10 metrics in all cross-domain query scenarios. This confirms the effectiveness of the SRM, which is used for local semantic mining and FMT, which is used for feature space stabilization modules to enhance CDIR performance. R@5 metrics can be referred in appendix A.1.

To evaluate the performance of our framework on diverse cross-domain scenarios, we conducted experiments on the DSR dataset. The evaluation results are presented in Table 1. In the text-to-image retrieval task, the improvement is not significant. This is likely because existing pre-trained VLMs, such as CLIP already have mature image-text alignment capabilities, leaving limited room for further optimization in T→I retrieval. In contrast, for scenarios with larger domain gaps, S→I and M→I retrieval, LS-CLIP delivers substantial improvements gains 8.4% and 9.9% on R@1 over FreestyleRet, highlighting its effectiveness in CDIR.

To further validate LS-CLIP performance on our self-constructed CDIR-Flickr30k dataset in Table 2, we evaluated both its zero-shot and fine-tuned capabilities. For reproducibility, key testing parameters are consistent with the DSR experiment. The zero-shot here means that models were not fine-tuned on CDIR-Flickr30k and only used pretrained weights fine-tuned on DSR dataset for evaluation. The results show that CLIP+Ours achieves average improvements of 16.7% and 9.4%

Table 3: Zero-shot results (%) of our method (+Ours) and other baselines on DomainNet dataset.

| Methods | Clipart→Real | | Infograph→Real | | Painting→Real | | Quickdraw→Real | | Sketch→Real | | Average | |
|---|---|---|---|---|---|---|---|---|---|---|---|---|
| | P@50 | P@100 | R@50 | P@100 | P@50 | P@100 | P@50 | P@100 | P@50 | P@100 | P@50 | P@100 |
| CLIP | 47.9 | 37.93 | 33.92 | 29.47 | 7.13 | 5.51 | 47.40 | 40.72 | 44.54 | 36.92 | 36.18 | 30.11 |
| CLIP+Ours | 53.49 | 45.20 | 21.47 | 18.66 | 11.97 | 9.92 | 47.03 | 41.05 | 48.42 | 41.75 | 36.48 | 31.32 |

Table 4: Zero-shot results on Image-text contrastive datasets.

| Methods | Flickr30k | | | | | | MSCOCO | | | | | |
|---|---|---|---|---|---|---|---|---|---|---|---|---|
| | I→T | | | T→I | | | I→T | | | T→I | | |
| | R@1 | R@5 | R@10 | R@1 | R@5 | R@10 | R@1 | R@5 | R@10 | R@1 | R@5 | R@10 |
| CLIP | 84.5 | 96.6 | 98.7 | 64.1 | 86.9 | 91.8 | 56.3 | 78.8 | 87.0 | 36.1 | 60.9 | 71.1 |
| CLIP + Ours | 87.2 | 98.5 | 99.4 | 74.4 | 93.6 | 96.3 | 63.3 | 84.7 | 91.3 | 45.7 | 72.4 | 81.7 |

Table 5: Zero-shot results on RSTPReid benchmark.

| Methods | R@1 | R@5 | R@10 | mAP@10 |
|---|---|---|---|---|
| CLIP | 12.9 | 31.5 | 42.1 | 9.6 |
| CLIP+DSR | 15.2 | 36.4 | 47.6 | 13.2 |
| CLIP+CDIR-Flickr30k | **16.8** | **39.8** | **54.1** | **13.9** |

in the average metrics R@1 and mAP@10, respectively, over FreestyleRet. This result confirms the strong generalization ability of LS-CLIP in unseen cross-domain scenarios. We then integrated LS-CLIP into three baseline models and evaluated their fine-tuned performance on CDIR-Flickr30k. In particular, the most significant gains appear in S→I and M→I tasks, highlighting the advantage of LS-CLIP in extracting semantic information from cross-domain images with large domain gaps.

### 4.3.2 Zero-shot Image Retrieval

**Cross-Domain Image Retrieval** In order to validate that the model's strong performance on the fully synthetic CDIR-Flickr30k benchmark will translate to real-world, human-generated cross-domain data, we set up a zero-shot experiment on DomainNet Peng et al. (2019) as shown in Table 3. Results show that our model outperforms the base model in retrieval on human-generated cross-domain data, demonstrating its strong understanding of the generality and human perceptibility of cross-domain semantics. For more method comparisons be referred in appendix A.3

**Text-based Image Retrieval** To evaluate the generalizability of the LS-CLIP, we set up a zero-shot experiment on the Flickr30k Young et al. (2014) and MSCOCO Lin et al. (2014) dataset. The results are shown in Table 4. First, LS-CLIP effectively improves retrieval recall in both tasks and datasets, with the most prominent gain 10.3% R@1 in Flickr30k's I→T task. Second, LS-CLIP preserves the model's zero-shot generalization while boosting recall, which is attributed to the FMT module's role in stabilizing the feature space and avoiding overfitting to specific domains. Generally, LS-CLIP boosts semantic understanding while retaining strong cross-modal generalization.

**Dataset validation** To verify the effectiveness of our CDIR-Flickr30k dataset and validate the fine-grained semantic understanding ability of the model. We selected the RSTPReid benchmark Zhu et al. (2021) as the evaluation dataset. This choice is justified by the characteristics of RSTPReid. It includes various fine-grained descriptions of human-related aspects such as clothing style. The results in Table 5 show that the model finetuned on our dataset achieves the best performance in retrieval tasks, and show that our dataset helps the model better understand cross-domain semantics.

### 4.3.3 Ablation Study

We also set up an ablation study to explore the impact of different modules on the model, which were tested on the DSR and the CDIR-Flickr30k. The test conditions were consistent with the previous test experiments. The results are shown in Table 6. Reconstructing the ViT features of patches, the SRM module enables the model to learn local regional semantic representations, thus significantly improving local semantic retrieval performance when incorporated alone. The FMT module alse aligns local features in the cross-domain, amplifying this advantage. The results highlight the effectiveness of SRM and FMT in improving the ability of the model on CDIR.

Figure 4: Visualization of the experimental results on the DSR (left) and the CDIR-Flickr30k (right).

Table 6: Results of the ablation study of LS-CLIP on DSR and CDIR-Flickr30k datasets.

| Methods | DSR | | | | | | | | CDIR-Flickr30k | | | | | | | | | |
| | T→I | | S→I | | C→I | | M→I | | T→I | | S→I | | C→I | | M→I | | O→I | |
| | R@1 | R@5 | R@1 | R@5 | R@1 | R@5 | R@1 | R@5 | R@1 | R@5 | R@1 | R@5 | R@1 | R@5 | R@1 | R@5 | R@1 | R@5 |
|---|---|---|---|---|---|---|---|---|---|---|---|---|---|---|---|---|---|---|
| CLIP* | 80.7 | 98.8 | 34.9 | 64.4 | 51.7 | 83.0 | 64.2 | 84.6 | 97.8 | 99.9 | 46.3 | 70.3 | 72.3 | 92.2 | 51.0 | 74.5 | 95.8 | 98.0 |
| +SRM | 81.8 | 99.0 | 76.3 | 96.0 | 67.0 | 95.6 | 91.7 | 98.6 | 97.9 | 99.8 | 81.3 | 91.8 | 76.6 | 90.9 | 78.9 | 84.3 | 97.9 | 99.6 |
| +SRM+FMT | 82.4 | 98.9 | 79.1 | 97.2 | 68.3 | 95.7 | 93.8 | 99.3 | 97.9 | 99.9 | 87.1 | 97.5 | 81.7 | 96.5 | 86.4 | 96.4 | 98.9 | 99.9 |

Table 7: Comparison of model parameter and inference speed on per batch. Our method is computationally efficient from the model parameters and inference speed aspects.

| Method | Parameters(M) | Speed(ms) |
|---|---|---|
| FreestyleRet-CLIP | 454 | 308 |
| CLIP | 408 | 19 |
| CLIP+Ours | 410(**+2**) | 21(**+2**) |

### 4.3.4 COMPUTATION COMPARISON

To verify the lightweight nature of our LS-CLIP and its ease of integration with existing retrieval models, we analyzed the computational complexity of LS-CLIP compared to other baselines. Table 7 presents a statistical analysis of the model parameters and inference time of per batch for our LS-CLIP framework and other baselines. In Table 7, compared to FreestyleRet, LS-CLIP is more lightweight in terms of both model parameters and inference speed. Relative to CLIP, LS-CLIP maintains efficient deployability while only slightly increasing model parameters and inference time.

### 4.3.5 VISUALIZATION

As shown in Fig.4, we visualize the retrieval results of CLIP Radford et al. (2021), FreestyleRet (abbreviated as FRet) Li et al. (2024) and our LS-CLIP model on DSR Li et al. (2024) dataset and CDIR-Flickr30k dataset. On DSR, we compared models' retrieval results post-training. On CDIR-Flickr30k, we evaluated their zero-shot retrieval after DSR training. In Fig.4, our method can retrieve images of people riding horses or people on boats on the water. It can also retrieve global scene images based on local ground information, and retrieve images matching the scene and characters from blurry pictures. Since our proposed method can not only preserve the global features of CLIP but also effectively mine its local features, thereby achieving better alignment with the local semantic information in images. The results in the figure show that our method has a better understanding of both local and global semantics compared to other models.

## 5 CONCLUSION

In this study, we propose LS-CLIP, a lightweight method suitable for various multimodal network patterns. LS-CLIP fully leverages the existing capabilities of mature multimodal models to excavate more latent knowledge to enhance the model's own abilities. Strengthens the model's performance in downstream tasks of cross-domain image retrieval without compromising its generalization ability. In addition, we also introduce a newly constructed CDIR dataset named CDIR-Flickr30k, in order to provide convenience to subsequent researchers.

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

## A  EXPERIMENTAL SUPPLEMENTARY MATERIALS

### A.1  RESULTS FOR R@5

In this section, we additionally supplemented the R@5 metrics in Table 1 and Table 2 for evaluation. The evaluation results are shown in Table 8 and Table 9. It can be seen from the evaluation results that under the R@5 metrics, LS-CLIP can still achieve excellent results.

Table 8: Results (%) comparison between our method (+Ours) and other baselines on DSR.

| Methods | T → I | S → I | C → I | M → I | Average |
|---|---|---|---|---|---|
| | R@5 | R@5 | R@5 | R@5 | R@5 |
| FreestyleRet | 97.9 | 95.2 | 95.3 | 96.6 | 96.3 |
| CLIP* | 98.8 | 64.4 | 83.0 | 84.6 | 82.7 |
| CLIP+Ours | 98.9 | 97.2 | 95.7 | 99.3 | 97.8 |
| FG-CLIP* | 99.2 | 71.2 | 91.2 | 88.7 | 87.6 |
| FG-CLIP+Ours | 99.2 | 95.0 | 94.9 | 98.9 | 97.0 |
| Siglip2* | 94.5 | 87.2 | 93.5 | 91.7 | 91.7 |
| Siglip2+Ours | 94.7 | 94.2 | 94.1 | 94.7 | 94.4 |

### A.2  EVALUATION OF VIT-B-32

To verify that LS-CLIP is a flexible and pluggable module, we also conducted corresponding retrieval experiments on ViT-B-32 and compared it with the baseline. The experimental settings can refer to the above experiments. The experimental results are shown in Table 10. Judging from the experimental results, our model can bring improvements not only on Vit-L-14 but also on Vit-B-32.

Table 9: Results (%) of our method (+Ours) and other baselines on CDIR-Flickr30k.

| Methods | T → I | S → I | C → I | M → I | O → I | Average |
|---|---|---|---|---|---|---|
| | R@5 | R@5 | R@5 | R@5 | R@5 | R@5 |
| FreestyleRet^ | 99.3 | 80.1 | 87.4 | 84.2 | 99.9 | 90.2 |
| CLIP+Ours^ | 99.9 | 95.2 | 95.7 | 95.2 | 99.8 | 97.2 |
| CLIP* | 99.9 | 70.3 | 92.2 | 74.5 | 98.0 | 87.0 |
| CLIP+Ours | 99.9 | 97.5 | 96.5 | 96.4 | 99.9 | 98.0 |
| FG-CLIP* | 99.9 | 77.3 | 95.2 | 75.0 | 99.6 | 89.4 |
| FG-CLIP+Ours | 99.9 | 98.8 | 96.2 | 98.6 | 99.9 | 98.7 |
| Siglip2* | 99.9 | 89.8 | 96.1 | 79.4 | 99.9 | 93.0 |
| Siglip2+Ours | 99.9 | 97.1 | 96.3 | 93.0 | 99.9 | 97.2 |

Table 10: Results(%) of LS-CLIP with ViT-B-32 on the DSR and CDIR-Flickr30k.

| Methods | T→I | | S→I | | C→I | | M→I | | O→I | |
|---|---|---|---|---|---|---|---|---|---|---|
| | R@1 | R@5 | R@1 | R@5 | R@1 | R@5 | R@1 | R@5 | R@1 | R@5 |
| Diverse-Style Retrieval Dataset | | | | | | | | | | |
| CLIP(ViT-B)* | 79.4 | 98.3 | 45.3 | 79.0 | 61.0 | 92.5 | 76.2 | 95.4 | / | / |
| w/ LS-CLIP | 81.0 | 98.8 | 58.4 | 87.0 | 62.2 | 93.2 | 78.0 | 95.5 | / | / |
| CDIR-Flickr30k Dataset | | | | | | | | | | |
| CLIP(ViT-B)* | 95.6 | 99.7 | 33.9 | 67.5 | 65.5 | 91.9 | 45.0 | 78.2 | 96.5 | 98.9 |
| w/ LS-CLIP | 96.3 | 99.6 | 57.8 | 86.5 | 68.8 | 92.0 | 64.6 | 89.2 | 96.3 | 99.0 |

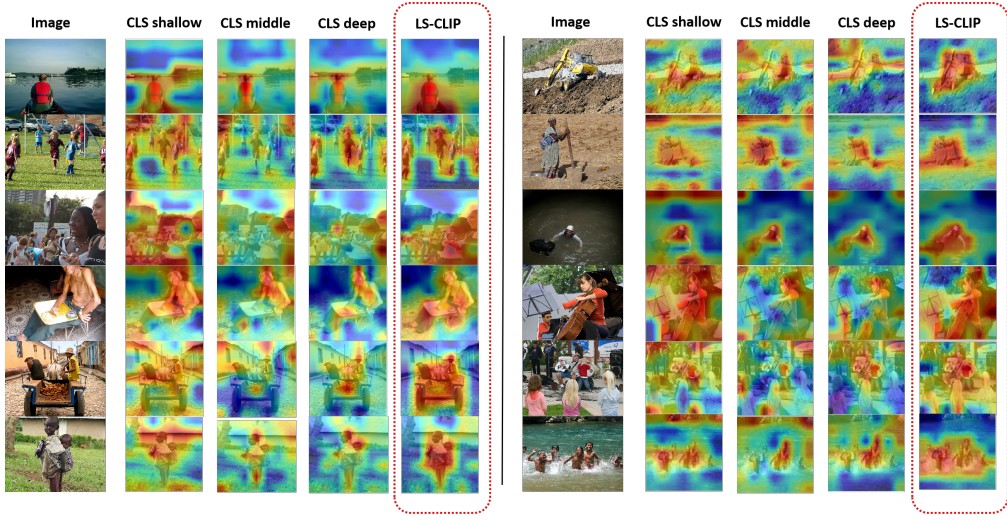

Figure 5: Visualization of attention weights. We presented the attention distribution of the CLS token in CLIP at different depths of the model, and compared it with that of LS-CLIP.

## A.3 ZERO-SHOT ON DOMAINNET DATASET

Built on algorithm-generated images, as shown in Table 11 our method still outperforms the base model on human-curated DomainNet. This shows that it is not dependent on specific data sources, can adapt to real human-generated cross-domain data, and thus has strong generalization ability. Furthermore, superior performance in zero-shot scenarios indicates that our method has a more accurate understanding of cross-domain semantics. Even without prior exposure to domain-specific

Figure 6: Additional visualization of attention weights. It can be referred to Figure 5 for details.

Table 11: Zero-shot results (%) of our method (+Ours) and other baselines on DomainNet dataset.

| Methods | Clipart→Real | | Infograph→Real | | Painting→Real | | Quickdraw→Real | | Sketch→Real | | Average | |
|---|---|---|---|---|---|---|---|---|---|---|---|---|
| | P@50 | P@100 | R@50 | P@100 | P@50 | P@100 | P@50 | P@100 | P@50 | P@100 | P@50 | P@100 |
| FG-CLIP | 44.37 | 37.60 | 36.26 | 32.65 | 2.31 | 2.01 | 52.93 | 47.46 | 40.25 | 35.54 | 35.22 | 31.05 |
| FG-CLIP+Ours | 51.78 | 45.07 | 27.16 | 24.52 | 5.97 | 5.16 | 51.47 | 46.37 | 47.99 | 42.94 | 36.87 | 32.81 |

domain data, it can still achieve effective transfer—further verifying the rationality and superiority of the method design. Finally, the result shows that our method still achieves improvements across different base models, which demonstrates its effectiveness.

### A.4 ANALYSIS OF ATTENTION MAP

In Figures 5 and 6, we visualized the attention distribution of the CLS token in the shallow, middle, and deep layers of CLIP, and compared it with the attention distribution of LS-CLIP on the image content. The shallow, middle and deep layers of the CLS token selected the results of the first, middle, and last layers of ViT, respectively. Judging from the results, our model can capture the main content in the image. The attention distribution is relatively concentrated. In terms of the attention distribution, the noise points of the attention weights of LS-CLIP are fewer, and it is more consistent with the contour of local objects, which is also consistent with our analysis results. By analyzing the attention of the shallow, middle and deep layers of the CLS token, we can know that in the shallow module of ViT, the model is more sensitive to local information, but there are still obvious noise points, and this phenomenon is well solved in LS-CLIP. As the number of layers increases, the CLS token tends to extract high-level semantic information, while the extraction of local information is no longer significant.

## B DATASETS

**Dataset Generation** Due to the high diversity of image descriptions provided by human annotators in the Flickr30k dataset, which can comprehensively describe the contents in images, we extended the Flickr30k dataset to CDIR-Flickr30k by introducing different styles and local objects of images.

- Text: The same as the Flickr30k dataset, each image has five captions.
- Sketch: The first caption of the image was selected as the text prompt for the generation of the sketch. The Pidinet Su et al. (2021) method was used to generate the image sketch.

- Mosaic images: The original images were converted into low-resolution images via down-sampling as queries with a scaling factor of 17.

- Cartoon: AnimateDiff Guo et al. (2023) was used to generate the corresponding artistic-style images.

- Objects: In practical application scenarios, global image retrieval by target is also common. Thus, GroundingDINO Liu et al. (2024) was used to extract local target images based on caption descriptions.

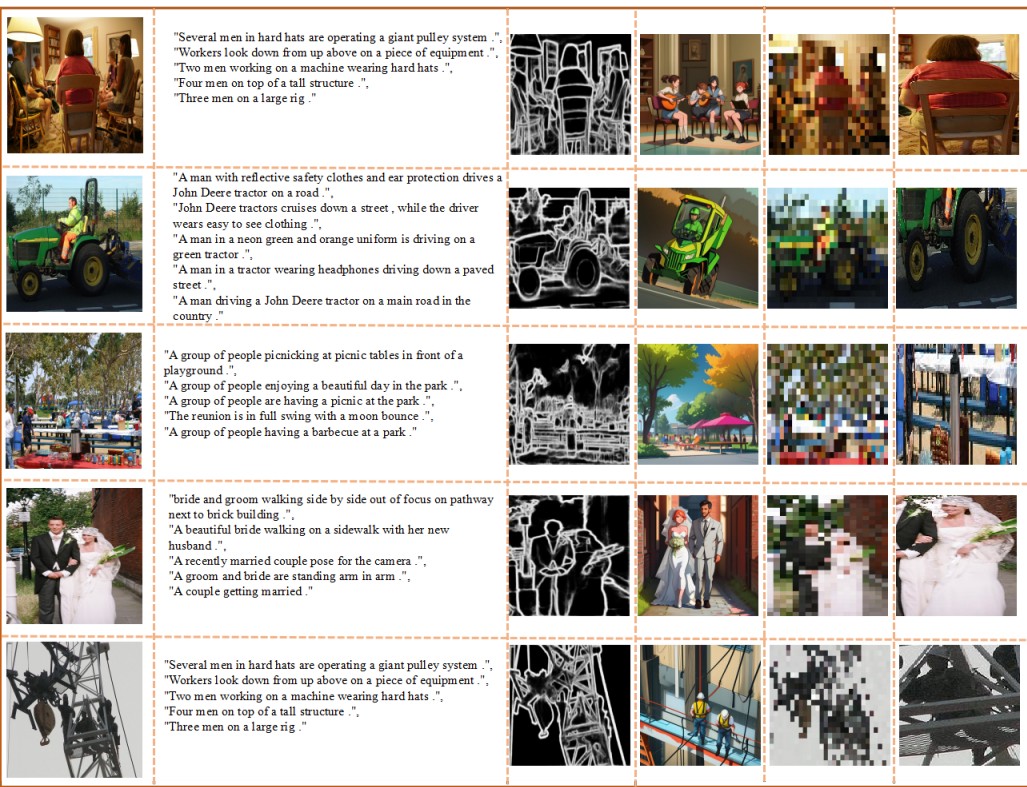

Figure 7: Samples of CDIR-Flickr30k. From left to right they are image, caption, sketch, cartoon image, mosaic image and object.

**Dataset Advantages** Compared with existing datasets, CDIR has the following advantages.

- **Compare with LAION Jia et al. (2021)** CDIR-Flickr30k offers stronger fine-grained capabilities. It refines both images and text. For images it expands the original image pool and incorporates more similar images. For text it breaks down coarse-grained sentences into fine-grained ones. This refinement greatly facilitates fine-grained semantic understanding. In contrast LAION, despite its large scale, has relatively coarse granularity in both text and images. CDIR-Flickr30k is more suitable for evaluating specific tasks. It is an improved dataset designed for tasks such as image-text retrieval. It provides more targeted support for assessing model performance in specific tasks like cross-modal fine-grained retrieval. LAION, on the other hand, focuses more on broad applications such as multi-modal pre-training.

- Compare with COCO Lin et al. (2014) CDIR-Flickr30k achieves higher relevance between images and text. The relevance between its text descriptions and images has been further optimized. In fine-grained retrieval tasks its text can describe image content more accurately. COCO, by comparison, falls slightly short in terms of fine-grained text descriptions. CDIR-Flickr30k demonstrates better cross-domain adaptability. As a cross-domain dataset improved based on Flickr30k it performs better in cross-domain tasks such as image-text

matching. COCO mainly focuses on tasks like common object detection segmentation and general image description so it is less adaptable in cross-domain scenarios.

- Compare with DSR Li et al. (2024) CDIR-Flickr30k contains a larger number of image-text pairs. DSR mainly includes 10,000 natural images along with four corresponding retrieval styles. CDIR-Flickr30k has an advantage in the scale of both images and text. This allows it to provide more data samples for model training and evaluation. CDIR-Flickr30k has richer text descriptions. Its text descriptions are refined fine-grained sentences which are more detailed and accurate. DSR's text descriptions are mainly designed to match its four retrieval styles so they are less rich and accurate in comparison. In addition the CDIR dataset also proposes an object-based retrieval task. This task can not only verify language retrieval at different granularities but also verify visual retrieval at different granularities.

## C  DESCRIPTION OF LLM USAGE

In this paper, we use and only use the Doubao LLM and the DeepSeek LLM for the paper grammar modification and word error correction, including parts such as the Introduction and Related work.

