# OpenReview forum: "LS-CLIP: Autoencoder-Based Mining of CLIP's Inherent Local Semantics in Cross-Domain Image Retrieval"
_ICLR.cc/2026/Conference — ICLR 2026 Conference Withdrawn Submission_

### Official Review · Reviewer_mWzY · 2025-10-19

**Soundness:** 3
**Presentation:** 2
**Contribution:** 2
**Rating:** 4
**Confidence:** 3

**Summary:**

This paper introduces LS-CLIP, a lightweight autoencoder-based module designed to mine local semantics from CLIP's patch features for Cross-Domain Image Retrieval (CDIR). The method comprises two main components: a Semantic Reconstruction Module (SRM) to reconstruct patch features for local detail mining and a Feature Moment Transfer (FMT) mechanism, inspired by style transfer, to enhance feature stability and generalization. The authors also contribute a new synthetic CDIR benchmark, CDIR-Flickr30k, which includes an object-to-image retrieval task .

**Strengths:**

1. **Plug-and-Play Efficacy**: The proposed module is lightweight, adding only 2M parameters , and demonstrates consistent performance improvements as a plug-and-play addition to various baselines (CLIP, FG-CLIP, Siglip2) .
2. **Resource Contribution**: The new CDIR-Flickr30k dataset introduces a novel object-to-image retrieval task, providing a new resource for evaluating multi-granularity visual retrieval.

**Weaknesses:**

1. **Limited Novelty**: The methodological contribution is incremental. The SRM is a standard autoencoder applied to patch features, conceptually similar to MAE , while the FMT module is a direct adaptation of the AdaIN mechanism from style transfer, used here as a regularizer.
2. **Weak Justification for FMT**: The paper posits that the complex FMT perturbation enhances generalization. However, it lacks a clear justification for why this specific moment-transfer operation is superior to simpler and more common feature-space regularization techniques (e.g., additive noise).
3. **Synthetic Benchmark Concerns**: All cross-domain queries in the CDIR-Flickr30k dataset are generated by other models (e.g., Pidinet, AnimateDiff, Grounding DINO). This raises a significant concern that the model is learning to invert the specific artifacts of these generators rather than learning a general, human-perceived understanding of cross-domain semantics.

**Questions:**

1. What is the clear justification for using the complex, style-transfer-based FMT mechanism for regularization over simpler, more established methods like dropout or noise injection in the feature space?

2. To properly ablate the method, can the authors provide results for a Base + FMT configuration (i.e., without the SRM module)?

3. How can the authors validate that the model's strong performance on the fully synthetic CDIR-Flickr30k benchmark will translate to real-world, human-generated cross-domain data, and is not just an artifact of learning to invert the data generators?

---

> ### Author Response · Authors · 2025-11-18
>
> Thank you for your reply. The following are the answers to the related questions:
> 1. The use of style transfer for feature moment migration aims to make the model focus more on semantic information rather than style information. Therefore, we supervise the style information by adding moment migration, enabling the model to be desensitized to this process.
> 2. In the settings of different ablation experiments, we consider that Experiments 1 & 2, as well as Experiments 2 & 3, are pairwise control experiments. They satisfy the principle of single variable and can support the experimental conclusions. However, we are willing to refer to your suggestions for appropriate experimental design.
> 3. Regarding the model's performance in real-world application scenarios, we have conducted experimental verification using downstream tasks that are close to practical scenarios, such as image-text retrieval and ReID. We hope this addresses your concerns, and we will carefully consider your suggestions.
>
> Wish you all the best.

---

> > ### Comment · Reviewer_mWzY · 2025-11-21
> >
> > Thank you for your response. Considering other reviewers' comments, I would really like to see strong evidence from your data to counter some of our potentially inappropriate views. I am very open to seeing your additional experiments (for example, to address my third point of weakness, you could provide data showing why the current results reflect real - world applications), and I would be happy to increase my score in light of such evidence.

---

### Official Review · Reviewer_oaby · 2025-10-30

**Soundness:** 1
**Presentation:** 1
**Contribution:** 2
**Rating:** 0
**Confidence:** 5

**Summary:**

The paper introduces LS-CLIP, a lightweight and plug-in local semantic mining framework built on CLIP for cross-domain image retrieval (CDIR) tasks such as sketch-to-image, cartoon-to-image, and mosaic-to-image retrieval.

**Strengths:**

- Achieves substantial performance improvements in challenging cross-domain settings (e.g., sketch/photo, mosaic/photo) with minimal computational cost, advancing practical retrieval applications.

- Contribution breadth: Provides not only an effective method but also a new dataset (CDIR-Flickr30k) that broadens evaluation coverage and supports future research in multi-style retrieval

**Weaknesses:**

## General ##
The paper suffers from major issues in writing and presentation. Many claims are incorrect, insufficiently supported, ambiguous, or irrelevant. Below, I outline several specific examples illustrating these problems. In its current form, the paper is not yet suitable for publication, particularly not at a top-tier venue such as ICLR:

- Abstract clarity: The abstract is unclear and should be rephrased. It fails to convey the core ideas and goals of the paper, relying instead on vague technical statements. For example, phrases like "Through reconstructing the patch features of the Vision Transformer (ViT), SRM integrates global and local semantic information…" are too abstract and do not help the reader understand the module’s purpose. Similarly, the mention of "Feature Moment Transfer (FMT)" and "stability of the feature space" is confusing - it is not clear what is being reconstructed, what "moment transfer" means, or how it contributes to feature stability. Overall, the abstract leaves the reader more confused than informed.

- Introduction – unsupported claim: The statement "...among which Query-Based Image Retrieval (QBIR) is the most widely applied" is misleading and lacks evidence. The authors should provide a reliable reference to support this claim. As far as is generally known, text-to-image search (e.g., Google Images) has been the most widely used Image Retrieval form for many years, more than image-to-image search. This needs clarification or revision.

- Figure 1 explanation: the figure is poorly explained. CLIP does not include a "cross-attention" module but rather self-attention, so it is unclear what exactly is being visualized. Are the authors showing attention maps between the CLS token and patch tokens? What visualization method or tool was used? The mention of "attention weights" in line 91 adds to the confusion and requires clearer explanation.

- Figure 1 caption: The caption is ambiguous due to the use of green, yellow, and red colors both for frame borders and in the heatmaps. The color references should be clarified to avoid misinterpretation.

- Line 84 – missing context: The sentence "we designed an autoencoder-based adapter named Semantic Reconstruction Module (SRM) based on MAE He et al. (2022)" is insufficient. The authors should briefly explain what MAE (Masked Autoencoder) contributed or how it relates to their method, rather than simply citing it without context.

- Line 92 – unsupported statement: The claim "However, existing benchmarks are insufficient" needs stronger justification. The provided explanation that "FSCOCO only supports text or sketch queries" is not convincing, as FSCOCO therefore was not intended for cross-domain image-to-image retrieval. Likewise, stating that "FreestyleRet has low-quality images" is not an adequate reason. The authors should discuss other relevant datasets, such as "Sketchy" [1], and explain more concretely why current benchmarks fail to meet their goals.

- Conceptual inconsistency: The authors contradict themselves: at line 37, they define CDIR as "finding relevant images in one visual domain based on query images from another visual domain", but later (line 132), they describe "image-text datasets as the foundation of VLM-based CDIR". These two definitions are inconsistent and need reconciliation.

- Line 133 – inaccurate phrasing: The sentence "Datasets like LAION, COCO, and Flickr30K focus on contrastive text-image learning in the same domain, lacking support for cross-domain query types" is conceptually incorrect. Datasets do not "focus" on learning methods - they only provide data. Furthermore, the phrase "lacking support for cross-domain query types" is unclear, as these datasets are inherently image–text paired datasets, not designed image-to-image retrieval (or for object detection, for example). The authors should refine this statement to accurately describe the datasets’ limitations.

## Evaluation ##
- Limited dataset validation: The authors claim improvements in cross-domain retrieval tasks, yet most of their evaluation is conducted only on the proposed CDIR-Flickr30k dataset. To convincingly demonstrate the method’s general effectiveness, results should be reported on additional, well-established cross-domain benchmarks. Relying solely on a self-introduced dataset limits the credibility and generalizability of the findings.
- Missing key baseline: The paper frames the main task as image-to-image retrieval, yet uses CLIP, a model primarily designed for image–text alignment, as its core backbone. This choice is questionable. Models such as DINOv2, which are widely adopted for image-to-image tasks and often outperform text-aligned models in pure visual retrieval, represent a critical missing baseline. Including DINOv2 or similar self-supervised vision models would provide a fairer and more meaningful comparison.

- Unclear evaluation focus: In Section 4.3.2, the authors present zero-shot text–image retrieval results. However, the primary goal of this work is cross-domain image-to-image retrieval. It remains unclear whether LS-CLIP improves general image retrieval performance, or if its gains are confined to specific setups involving CLIP. The authors should clarify whether their approach enhances retrieval in general, or only when coupled with CLIP for cross-domain adaptation.


[1] Patsorn Sangkloy, Nathan Burnell, Cusuh Ham, and James Hays. 2016. The sketchy database: learning to retrieve badly drawn bunnies. ACM Trans. Graph. 35, 4, Article 119 (July 2016), 12 pages.

**Questions:**

See "weaknesses".

---

> ### Author Response · Authors · 2025-11-18
>
> Thank you for your reply. The following are the answers to the related questions:
> 1. CLIP includes an attention module and can naturally visualize the attention weight distribution among different patches. This is feasible, and we have also elaborated on the visualized content in the paper, such as the depth of network layers and the selection of patches.
> 2. If the relevant expressions have caused ambiguity for you, we will carefully review them in accordance with your suggestions.
>
> May you enjoy your life.

---

> > ### Comment · Reviewer_oaby · 2025-11-23
> >
> > I appreciate the authors response. However, my concerns remain unaddressed. Therefore, I will maintain my original score and hope the authors will consider above points in future revisions of this paper.

---

### Official Review · Reviewer_1qJo · 2025-11-01

**Soundness:** 3
**Presentation:** 3
**Contribution:** 2
**Rating:** 2
**Confidence:** 3

**Summary:**

The paper introduces a lightweight framework that enhances CLIP’s local semantic understanding for cross-domain image retrieval. It proposes two modules: a Semantic Reconstruction Module (SRM) that mines local patch features through self-supervised reconstruction, and Feature Moment Transfer (FMT) that improves feature stability via cross-domain moment alignment. To better evaluate multi-granularity retrieval, the authors also build a new dataset, CDIR-Flickr30k, containing diverse query types such as text, sketch, and object images. Experiments show that LS-CLIP consistently improves performance and generalization across multiple CLIP variants while maintaining low computational cost.

**Strengths:**

1. The paper identifies an important limitation of CLIP—its weak handling of local semantics—and attempts to address it in the context of cross-domain retrieval.
2. The proposed method (LS-CLIP) is lightweight and can be integrated into existing CLIP models without significant computational overhead.
3. The newly constructed CDIR-Flickr30k dataset broadens the evaluation perspective by including diverse query modalities such as sketch and object images.
4. Experimental results are clearly presented and demonstrate consistent, if moderate, improvements across multiple CLIP variants and datasets.

**Weaknesses:**

1. The overall contribution is incremental rather than fundamental; the method primarily refines feature representation without introducing new theoretical insights or a strong methodological innovation.
2. The technical novelty is limited, both the autoencoder reconstruction and moment alignment components are standard techniques reused with minimal adaptation.
3. The proposed CDIR-Flickr30k dataset lacks detailed description, justification, and validation, making it unclear whether it meaningfully challenges existing benchmarks.
4. The paper focuses heavily on quantitative results but lacks qualitative or interpretive analysis to demonstrate why LS-CLIP captures “local semantics” more effectively.
5. There is no convincing discussion of practical significance; the improvement margins are small and do not clearly justify the complexity of the additional modules.
6. The writing sometimes overstates the contribution, giving the impression of conceptual novelty where the actual advance is modest.

**Questions:**

Please see the weaknesses.

---

> ### Author Response · Authors · 2025-11-18
>
> Thank you for your reply. The following are the answers to the related questions:
> 1. The performance improvements and highlights of LS-CLIP have all been pointed out in the original paper, so they will not be repeated here. However, we respect all your opinions.
> 2. A detailed description of the CDIR-Flickr30k dataset can be found in Appendix B, and the positive effects brought by the new dataset can be referred to in Table 4 of the paper. We hope this will answer your questions.
> 3. For the qualitative analysis of LS-CLIP, please refer to Figures 1 and 4 of the paper as well as Appendix A.3. We hope this will be helpful to you.
>
> Wish you all the best.

---

> ### Comment · Reviewer_1qJo · 2025-11-20
>
> Thank you for your detailed explanation and for pointing me to the relevant parts of the appendix.  After re-reading the manuscript, I admit that some of the questions I raised were due to my negligence in the supplementary.  In particular, the extended description of the CDR-Flickr 30k dataset in Appendix B, as well as the additional qualitative visualizations in Figures 1, 4, and Appendix A.3, do address some of my earlier concerns about dataset construction and qualitative evidence. But I still have concerns about the novelty of the LS-CLIP method.  Reconstruction and moment alignment based on automatic encoders use standard techniques, the motivation of using them is not clear.  Nonetheless, I agree that the empirical results show consistent improvements across several search scenarios, and that the dataset has a positive contribution to evaluating cross-domain searches. Taking these clarifications into account, I adjusted my score upward to better reflect the integrity of the submission and the value of the empirical findings. Thank you!

---

### Author Response · Authors · 2025-12-03
**response**

We would like to express our special thanks to the reviewers for providing valuable comments on our manuscript. We have revised the paper in accordance with these comments and suggestions, and several modifications were made during the revision process. After this comprehensive revision and refinement, we believe the quality of the paper has been significantly improved.
The main revisions are as follows:

1)Regarding the paper’s expression issues: we carefully reviewed every comment from the reviewers and made revisions accordingly. Additionally, we invited several English professionals to polish the paper’s expression.

2)For technical revisions: in accordance with the reviewers' suggestions, we have corrected the erroneous descriptions and added supplementary explanatory analyses related to the technology.

3)Regarding the issue of innovation: First, we clarify the differences between our SMR and MAE. The core of the MAE method is to mask part of the input and reconstruct the masked regions; it enables the model to learn fine-grained image information through reconstruction loss and is mainly used for downstream tasks. By contrast, the core of our SMR is to compress high-dimensional features into low-dimensional features while avoiding the loss of key visual details. Notably, its decoder stage is only used for training and not employed during inference. As for the FMT module, our main innovation lies in stabilizing the distribution of cross-domain feature spaces through moment transfer. We are the first to propose the application of moment transfer to cross-domain image semantic alignment, which ultimately enhances the generalization ability of the model. Furthermore, we have explored enhancing fine-grained information within CLIP. This approach requires no additional annotated data and can capture fine-grained information with only a 2M parameter increase.

4)Regarding the dataset: To meet various user retrieval requirements, we have created CDIR-Flickr30k. Regarding the issue of innovation: First, we clarify the differences between our SMR and MAE. The core of the MAE method is to mask part of the input and reconstruct the masked regions; it enables the model to learn fine-grained image information through reconstruction loss and is mainly used for downstream tasks. By contrast, the core of our SMR is to compress high-dimensional features into low-dimensional features while avoiding the loss of key visual details. Notably, its decoder stage is only used for training and not employed during inference. As for the FMT module, our main innovation lies in stabilizing the distribution of cross-domain feature spaces through moment transfer. We are the first to propose the application of moment transfer to cross-domain image semantic alignment, which ultimately enhances the generalization ability of the model.

5)In order to validate that the model's strong performance on the fully synthetic CDIR-Flickr30k benchmark will translate to real-world, human-generated cross-domain data, we set up a zero-shot experiment on DomainNet as shown in Table 3. Results show that our model outperforms the base model in retrieval on human-generated cross-domain data, demonstrating its strong understanding of the generality and human perceptibility of cross-domain semantics. For more method comparisons be referred in appendix A.3. The result shows that our method still achieves improvements across different base models, which demonstrates its effectiveness.

---

### Note · Authors · 2025-12-09

**Comment:**

To the Program Committee of ICLR:

We, the corresponding author and all co-authors of the paper titled “LS-CLIP: Autoencoder-Based Mining of CLIP's Inherent Local Semantics in Cross-Domain Image Retrieval” (Submission ID: [17362]), hereby formally request the retraction of this submission.

Following the review process, we have carefully evaluated the feedback provided by the reviewers. Regrettably, one reviewer assigned an outright 0-point score without providing sufficient and reasonable justifications to support this evaluation. As authors committed to academic integrity and constructive exchange, we have not felt the equal respect that should underpin academic review, nor have we gained valuable insights or constructive suggestions that facilitate the improvement of our work. Given this circumstance, we collectively decide to withdraw the paper, as we believe such an unsubstantiated evaluation does not align with the principles of fair academic review.

We sincerely apologize for any inconvenience this may cause to the committee. We will further refine the work and consider resubmitting it in future ICLR cycles when appropriate.

Thank you for your understanding.

Sincerely,

On behalf of all co-authors

December 09, 2025

**Withdrawal Confirmation:**

I have read and agree with the venue's withdrawal policy on behalf of myself and my co-authors.